# Miniaturized Quad-Band Filter Design Using Substrate Integrated Coaxial Cavity

**DOI:** 10.3390/mi14020347

**Published:** 2023-01-30

**Authors:** Chung-I G. Hsu, Wanchu Hong, Mingchih Chen, Wei-Chun Tu, Gwan-Wei Su, Min-Hua Ho

**Affiliations:** 1Department of Electrical Engineering, College of Engineering, National Yunlin University of Science and Technology, Yunlin 64002, Taiwan; 2Department of Electronic Engineering, College of Engineering, National Changhua University of Education, Chunghua 50007, Taiwan; 3Graduate Institute of Business Administration, College of Management, Fu Jen Catholic University, New Taipei City 24205, Taiwan; 4Universal Microwave Technology, Inc., Keelung 20647, Taiwan; 5Wistron Corporation, Taipei 11469, Taiwan

**Keywords:** quad-band filter (QBF), substrate integrated coaxial cavity (SICC), coaxial cavity mode, circuit miniaturization

## Abstract

We propose a miniaturized quad-band filter (QBF), designed using substrate-integrated coaxial cavities (SICCs). The employed SICC structure consists of two vertically stacked substrates with a large circular patch embedded in between. The embedded patch is segmented unevenly into four pieces, which are shorted to the cavity’s bottom wall through one or two blind vias. This SICC structure exhibits four independently controlled resonances with frequencies much lower than the frequency of its conventional SIW cavity counterpart, thus achieving size reduction. A sample quad-band filter is designed and fabricated for experimental measurement. Reasonably good agreement between measured and simulated data is observed.

## 1. Introduction

The immense multiband-operation demand for today’s wireless communication systems requires multiband-function components. Among them, multiband filters, such as dual-, tri-, and quad-band, play essential roles in assuring high-quality performance by confining the system’s operation to allocated bands. In these multiband bandpass filters (BPFs), quad-band filters (QBFs) are the most difficult to design and have gained more and more attention in commercial mobile communication systems recently. In the past, multiple microstrip or ring resonators have been employed to build QBFs [1,2,3] because of the advantages of low cost and high circuit diversity. However, they share common deficiencies, such as low Q characteristics, low power handling capacity and low noise immunity to the outside environment. In [1], two distinct square rings generate two pairs of perturbed degenerate modes to build the QBF’s four passbands. The two square rings are vertically arranged to save circuit space. In [2], the presented QBF is comprised of four stepped-impedance resonators (SIRs), two of which generate the first and third passbands and the other two of which produce the second and fourth passbands. The drawback to this design is that high-quantity planar resonators in a close arrangement might complicate the design procedure and occupy a large circuit area. In [3], the QBF is built by complex SIRs loaded with multiple open stubs, resulting in an overwhelmingly complicated design procedure. This design also needs a considerable circuit area.

Decades ago, the substrate integrated waveguide (SIW), a waveguide-like structure, was built using printed circuit board (PCB) technology [4] to alleviate the noise immunity problem and to increase the circuit’s power handling capability and Q value. The SIW’s enclosure structure grants itself the advantages of the waveguide and still preserves the flexibility of PCB-based planar circuitry, such as the microstrip, slot-line, strip-line and coplanar waveguide. Later, SIW filters were extensively investigated; however, only limited QBFs were reported [5,6]. In [5], the ingenious QBF designed using an SIW loaded with four complementary split-rings (CSRs) in various forms successfully meets the quad-band requirement, and the signal selectivity enhancement is achieved by introducing transmission zeros (TZs) between adjacent passbands. Nevertheless, the CSRs might incur radiation losses and hinder high-frequency applications of the filter. In [6,7], the QBFs are built of multiple SIW cavities. The four bands are formed by splitting the two passbands caused by the cavity’s first and second resonances into four, where the band splitting is realized by dual-mode perturbation [6], or by implanting one or two TZs around the center of each original passband [7]. The drawback to this passband splitting design is lack of flexibility in passband allocations since the two split bands must be very close to each other and cannot be randomly deployed. In addition, the usage of many SIW cavities certainly leads to a considerable circuit area, not to mention that the size compactness of a single SIW cavity is also incomparable to the aforementioned PCB structures.

Recently, an SIW-cavity-like structure termed the substrate-integrated coaxial cavity (SICC) was proposed in [8], and later modified in [9], to design single-band BPFs. These SICC structures can effectively reduce the SIW cavity size and relieve the big-circuit-size deficiency incurred by the conventional SIW filter. The SICCs reported in [9] were subsequently reconfigured to design a dual-band BPF in [10] and a triband BPF in [11]. In [10], the embedded circular patch is evenly segmented into four pieces, where two oppositely deployed quarter-circular patches each shorted to the SICC’s bottom wall with one blind via are associated with the generation of the lower resonance frequency and the other two quarter-circular patches, each shorted to the SICC’s bottom wall with two blind vias, pertain to the generation of the higher resonance frequency. In [11], the embedded circular patch is evenly segmented into three pieces, each of which, when connected to the SICC’s bottom wall through one, two, or four blind vias, is associated with the generation of one specific resonance frequency of the triband BPF. In the two works mentioned, each segmented patch in conjunction with the connected blind vias can be regarded as an independently controlled resonator. The reason that these BPF structures can be miniaturized is that a single cavity houses multiple resonators, instead of just one as in [8,9].

In this paper, the structures reported in [10,11] are further improved to realize a QBF. In comparison, the structure presented here is more advantageous in design than those in [10,11] in the following respects. First, in terms of the equivalent LC resonance circuit, the two different resonators in [10] and the three resonators in [11] have the same capacitance but different inductances, since the segmented patches have the same area but are shorted to the ground with a different number of blind vias. The distinction of the resonance frequencies is controlled only through the inductance, a design method that may be regarded as lacking flexibility. If the same procedures reported in [10,11] are extended to construct a QBF, the fourth quarter-circular patch may need five or six blind vias to produce the needed fourth resonance frequency, causing the design process to become complicated. Here, by unevenly segmenting the circular patch into four fan-shaped patches, we can easily design a QBF with only one or two blind vias under the four fan-shaped patches. By changing the radius and subtended angle of the fan-shaped patch, we can control the capacitance; by changing the number and positions of the blind vias, we can control the inductance. Second, the four fan-shaped patches can be tuned to have different radii. As a result, the proposed design here is much more flexible than those presented in [10,11]. Third, control of the inductance for more than two blind vias is much more difficult than that for only one or two blind vias. This is because mutual magnetic coupling among multiple blind vias complicates the resultant effective inductance. The simulation in obtaining an appropriate effective inductance becomes progressively more time-consuming if more blind vias are added. Fourth, adjacent fan-shaped patches are separated by fan-shaped slits, a strategy that is easier for tuning structural dimensions in simulation as compared with straight slits employed in [10,11]. In short, the proposed design here is more robust than those presented in [10,11]. Reasonably good agreement between the measured and simulated data can be observed for the designed QBF.

## 2. Filter Design and Sample Results

In Figure 1a–d, we show the proposed SICC QBF circuit structure and layouts of the three metal layers. For distinction, the top, middle, and bottom metal layers are denoted by M1, M2, and M3, respectively, as indicated in Figure 1a. In between the three metal layers of each SICC are the two substrates with the same dielectric constant *ε_r_* and loss tangent tanδ. The thickness is *h*_1_ for the top substrate and *h*_2_ for the bottom. Note that metal layers M1 and M3 are the top and bottom walls of the cavity, respectively. Note also that the layout extent of metal layer M2 is smaller than the area of the rectangular SICC’s top/bottom wall, and hence metal layer M2 is called an embedded metal for convenience. The QBF is composed of two identical SICCs, each with four possibly distinct fan-shaped patches in metal layer M2. These fan-shaped patches are shorted to the cavity’s bottom wall (i.e., metal layer M3) through one or two blind vias (for convenience, denoted by shorting vias). An SMA is used to feed the SICC from the SICC’s bottom wall, where the SMA’s signal probe penetrates the cavity and reaches the top wall (i.e., metal layer M1). The two SICCs in a top-wall facing top-wall fashion are vertically bound, with metal layer M1 as their common top wall. Four distinct slots, also designed to be of fan shape, are etched on this common wall for coupling energy between the two stacked SICCs. Each fan-shaped patch together with its shorting via (or vias) can generate a rather independently controlled resonance for the corresponding passband and hence can be regarded as a resonator. Since four resonators are confined in the same rectangular cavity, miniaturization can then be effectively achieved in our proposed QBF design.

In the full-mode cavity of [9] and the half-mode cavity of [12], each cavity contains only a resonator and an empirical expression for the resonance frequency can be derived from an equivalent transmission-line model. In our design, although the four resonators are housed in the same cavity, each resonator can still be associated with one equivalent transmission-line model. Alternatively, based on the approximation that both substrates are very thin, each resonator can be regarded as a parallel LC resonant circuit between the fan-shaped patch and the ground and the resonance frequency is $f_{r}=1/(2\pi \sqrt{LC}).$ The capacitance is then $C=\varepsilon_{r}\varepsilon_{0}A_{p}(1/h_{1}+1/h_{2}),$ where *A_p_* is the area of the fan-shaped patch and $\varepsilon_{0}$ is the permittivity of vacuum. Unfortunately, unlike the cases where an analytic expression for the inductance *L* is available for a symmetric-via-loaded full circular patch in a cylindrical SICC [13] and where an empirical expression can be easily established for a symmetric-via-loaded full circular patch in a rectangular SICC [9], neither analytical nor empirical expressions for the asymmetric-via-loaded fan-shaped patch in the rectangular SICC in this paper can be found. Nevertheless, the empirical inductance expression $L=\mu_{0}h_{2}\ln(1.079W/R)/(2\pi )$ that can be established for [11] can still be used to estimate the position of the shorting via below a fan-shaped patch in the initial step of the design, where *W* is the width of the square cross-section of the rectangular cavity and *R* is the distance between the via and the center of the cavity’s square cross-section. This position can be fine-tuned in the subsequent simulations. In this paper, RT/Duroid substrates (*ε_r_* = 2.2 and tanδ = 0.0009) having the thicknesses of *h*_1_ = 0.254 and *h*_2_ = 1.58 mm are chosen to design the QBF. The structural dimensions of the QBF, simulated with the help of the software tool High-Frequency Structure Simulator (HFSS), are given in Figure 1. For distinction, the four patches with P1, P2, P3, and P4 denoted nearby in Figure 1c signify that their corresponding resonance frequencies are in ascending order. The P1 and P2 patches, having the same area but with a different number of shorting vias, are associated with the first and second resonances, respectively. They are oppositely deployed to minimize the mutual coupling in between.

In Figure 2a,b, we show the resonance frequency vs. the fan-shaped patch’s arc angle (*θ*) for several different values of the patch’s side dimension (*l*), with the inner radius of the patch fixed at 2 mm. For the patch with one shorting via, Figure 2a reveals that a larger patch leads to a lower resonance frequency because of the larger capacitance. When the fan-shaped patch is loaded with two shorting vias, the two corresponding inductances in parallel connection provide a smaller total inductance, causing the resonance frequency to increase, as shown in Figure 2b. The final structural dimensions for the four resonators, with the help of HFSS, are given in Figure 1. The unloaded Q-factors (*Q_u_*) of the resonators pertaining to the P1, P2, P3 and P4 patches in the SICC without the coupling slots are calculated to be 165, 184, 314 and 406, respectively. These *Q_u_* values are calculated using an in-house computer program developed according to the algorithm presented in [14]. Note that this algorithm is very accurate in estimating the *Q_u_* values, especially for narrow-band resonators.

Figure 3a–d and Figure 4a–d give the electric- and magnetic-field strengths on the top and bottom surfaces of metal layer M2, respectively, at the four passband frequencies. As shown in the figures, each resonance is dominated by only one corresponding patch. The patch and its shorting via(s) together with the cavity resonate at a quasi-coaxial-cavity mode. In Figure 4, the magnetic field circulates each of the two shorting vias in the same manner, which indicates that the vias’ currents are in the same direction. A fan-shaped slot of adequate size is embedded in the common wall of the two SICCs to provide the needed coupling in between.

In Figure 5a–b, we present the measured and simulated frequency responses for the proposed QBF given in Figure 1. Two transmission zeros (TZs) are generated between adjacent passbands. The TZs are due to the cancellation of the magnetic and electric couplings, a phenomenon that can also be found in [11]. The measured (simulated) mid-band frequencies of the passbands are 1.9 (1.86), 2.62 (2.6), 3.63 (3.64) and 4.63 (4.61) GHz with the corresponding fractional bandwidths (FBW) of 3.2% (3.2%), 2.8% (3.07%), 1.4% (1.37%) and 1.7% (2.1%), respectively. The measured and simulated minimum in-band insertion losses (from the low band to the high band) are 1.9 (1.86), 1.7 (1.6), 2.1 (2.3), and 2.2 (1.8) dB. Although independent control of the four passband center frequencies has been achieved, our design has not devised that of the four passband FBWs. To independently control the four FBWs, we need to control their corresponding external quality factors. For that purpose, we might be able to modify the shape of the annular-ring slot surrounding the SMA probe in the bottom wall. One possible way is to unevenly divide the annular-ring slot into four fan-shaped portions according to the ratios of the subtended angles of the four embedded patches. Then, the outer radii of the four fan-shaped portions are subsequently changed to obtain the desired external quality factors.

For circuit performance comparison, the circuit parameters of our work and some other SIW-related QBFs, all PCB based, are listed in Table 1. Note that *λ_d_* in Table 1 is the intrinsic wavelength in the dielectric medium at the first passband’s center frequency. In our design, if the thickness of the SICC’s top substrate is reduced to one-quarter of its original value, the occupied circuit area can be greatly reduced from 0.35 *λ_d_* × 0.35 *λ_d_* in Table 1 to approximately 0.18 *λ_d_* × 0.18 *λ_d_*. However, handling such a thin PCB in the SICC QBF fabrication process becomes quite challenging. As can be seen from this table, our proposed QBF possesses the widest relative upper stopband, which is very important for practical applications. In Figure 6a–b, we give photos of the SICC’s top wall (with coupling slots) and the bottom (furnished with an SMA).

## 3. Conclusions

In this paper, we present the design of a size-reduced SICC QBF. The desired four resonance frequencies can be obtained from one single size-reduced SICC, and hence, the proposed QBF consisting of two vertically stacked SICCs has led to excellent circuit-area efficiency. The overall circuit occupies an area of only 0.35 *λ_d_* × 0.35 *λ_d_*. The proposed SICC QBF with such an excellent size reduction and such a wide relative upper stopband is believed very suitable for commercial mobile communication applications.

## Figures and Tables

**Figure 1 micromachines-14-00347-f001:**
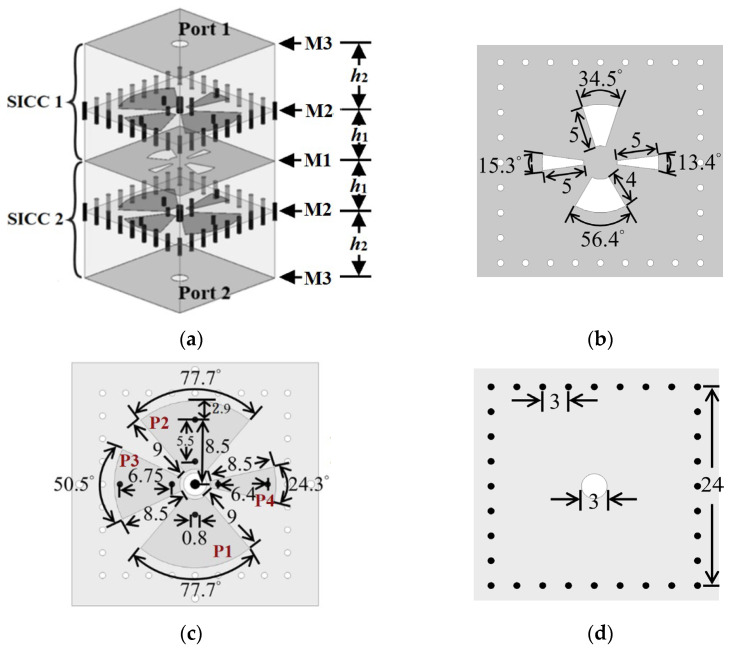
(**a**) The 3D view of the proposed QBF. (**b**) Metal layer M1 (with coupling slots embedded), (**c**) metal layer M2 (consisting of fan-shaped patches), and (**d**) metal layer M3 of each SICC.

**Figure 2 micromachines-14-00347-f002:**
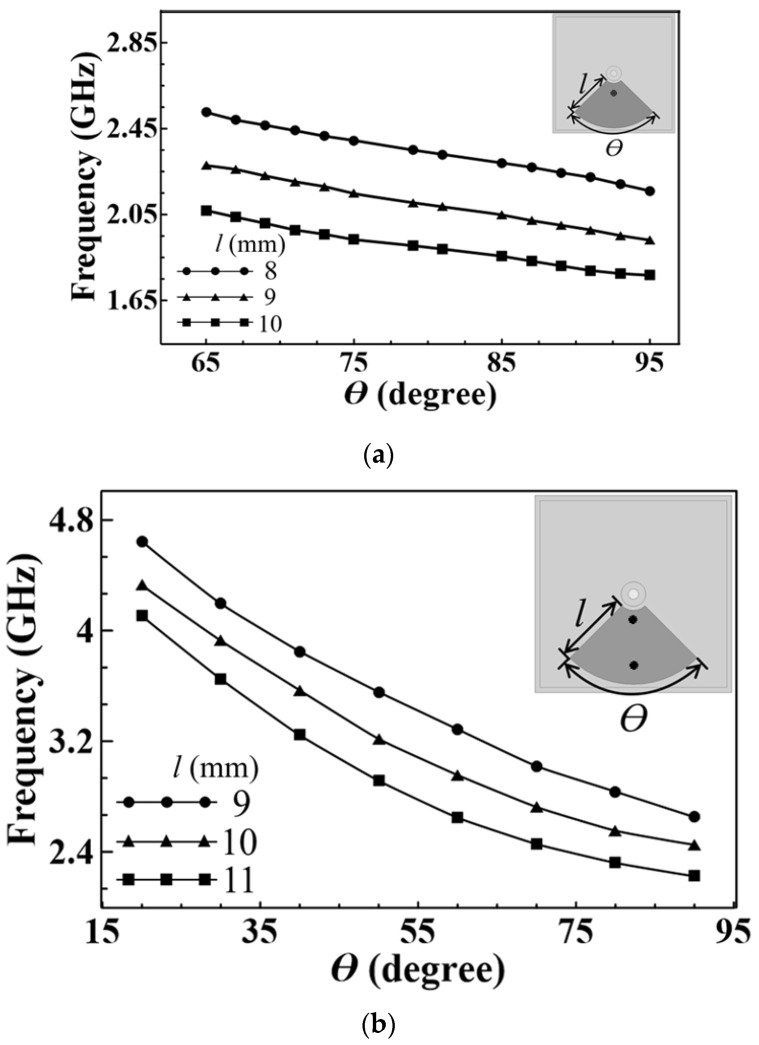
The resonance frequency as a function of the arc angle, *θ*, of the embedded fan-shaped patch for several side dimensions (*l*) with (**a**) one and (**b**) two shorting vias.

**Figure 3 micromachines-14-00347-f003:**
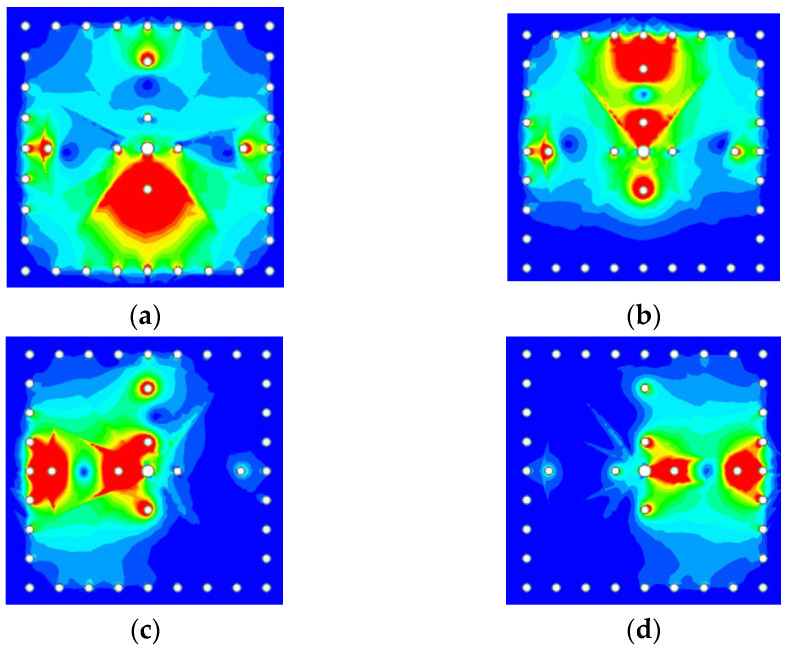
The electric field distribution on the top surface of metal layer M2 at the (**a**) first, (**b**) second, (**c**) third, and (**d**) fourth passband resonance frequencies.

**Figure 4 micromachines-14-00347-f004:**
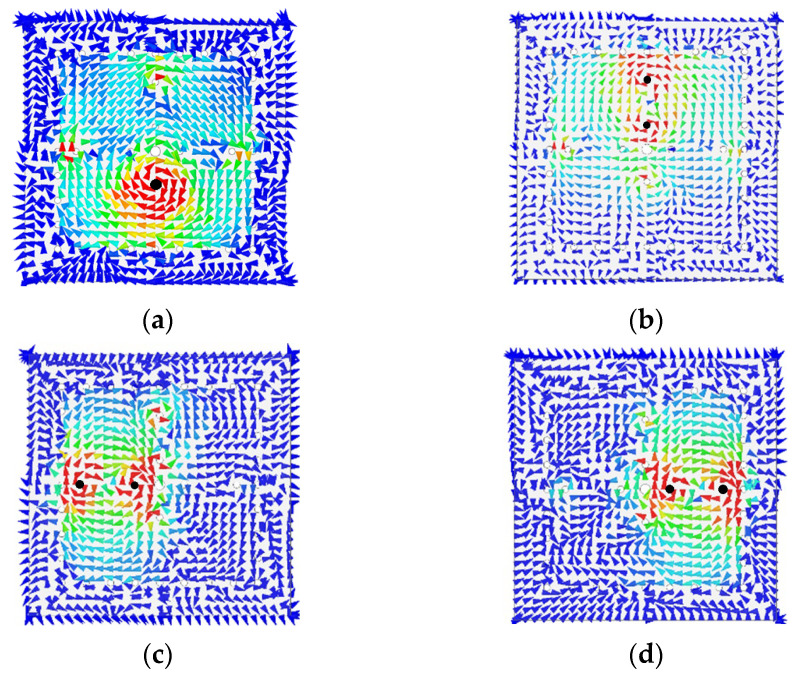
The magnetic field distribution on the bottom surface of metal layer M2 at the (**a**) first, (**b**) second, (**c**) third, and (**d**) fourth passband resonance frequencies.

**Figure 5 micromachines-14-00347-f005:**
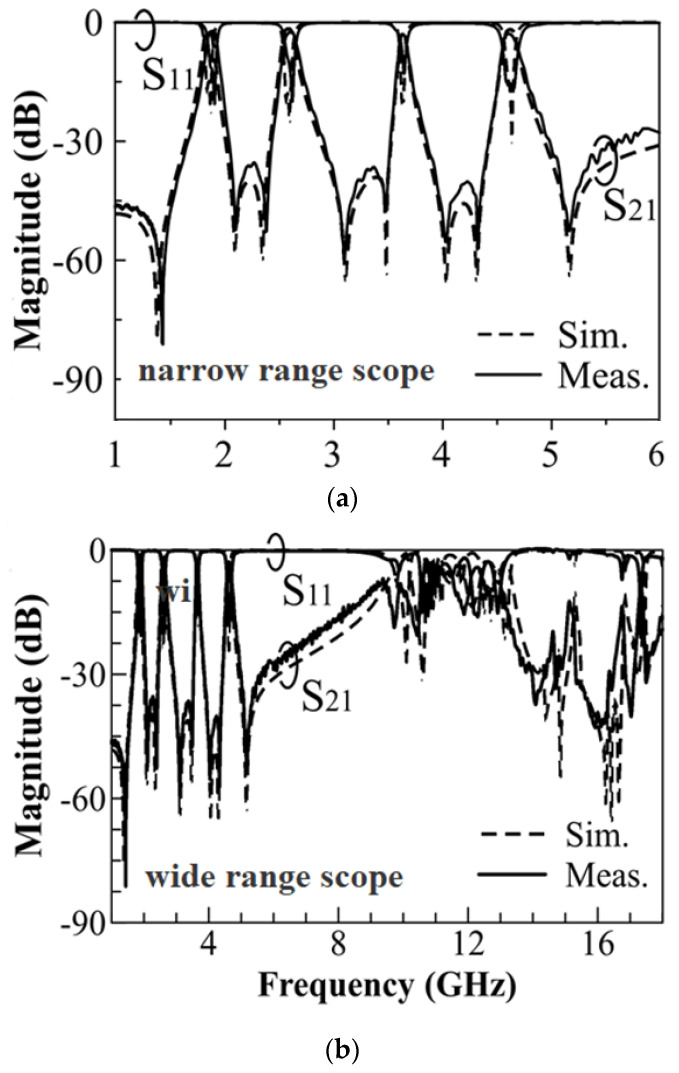
The measured and simulated frequency responses for the proposed QBF are shown in (**a**) narrow-range and (**b**) wide-range scopes.

**Figure 6 micromachines-14-00347-f006:**
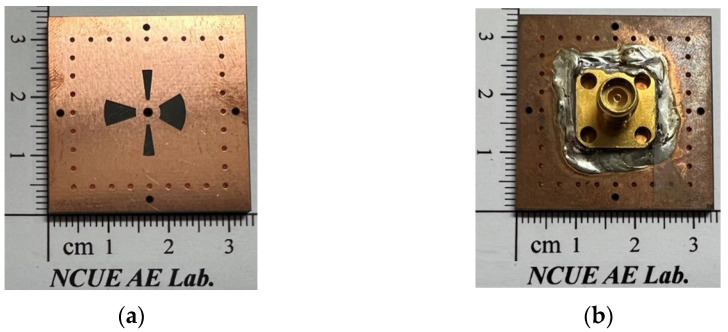
The photos of the experimental circuit’s SICC: (**a**) the coupling slots layer and (**b**) the bottom layer (attached by an SMA).

**Table 1 micromachines-14-00347-t001:** Circuit performance between our design and PCB-based SIW-related QBF references.

	*f*_1_/*f*_2_/*f*_3_/*f*_4_ (GHz)	FBW (%)	IL (dB)	Order	Size (*λ_d_* × *λ_d_*)	USBW |S_21_| <−20 dB
[5]	3.82/5.02/	7.2/1.58/	1.77/3.61/	2	0.26 × 0.22	NA
	6.12/9.07	1.78/1.34	3.47/4.47			
[6]	3.36/3.95/	6.9/8.6/	1.39/1.25/	2	0.31 × 0.71	~0.77 *f*_1_
	6.18/7.06	5.9/2.4	1.53/2.45			
[7]	11.5/12.5/	1.43/1.42/	1.33/1.22/	2	2.15 × 1.27	~0.065 *f*_1_
	14.7/15.2	1.14/1.0	1.43/1.53			
This work	1.9/2.62/	3.2/2.8/	1.9/1.7/	2	0.35 × 0.35	1.23 *f*_1_
	3.63/4.63	1.4/1.7	2.1/2.2			

USBW: abbreviation for upper stopband BW.

## Data Availability

The data presented in this study are available on request from the corresponding author.

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
