# Peer review of "Miniaturized Quad-Band Filter Design Using Substrate Integrated Coaxial Cavity"

_micromachines, 2023, doi:10.3390/mi14020347_

Round 1

Reviewer 1 Report

I don’t think the novelty of this work is sufficient for publication in this journal since the authors just extend their previously proposed resonator theory from dual-band [R1] and triple-band [10] BPFs development to a quad-band BPF design. In addition, the previous work [R1] from the same group must be cited:

[R1] Miniaturised substrate integrated waveguide cavities in dual-band filter and diplexer design, doi: 10.1049/iet-map.2019.0510

Author Response

Reply: We thank the reviewer for the comment. We have included Reference [R1] into the revised manuscript as Reference [10], and the original Reference [10] becomes Reference [11] in the revised manuscript. To emphasize our contribution, we have divided the last paragraph of the Introduction section in the original manuscript into two paragraphs. The first paragraph addresses the evolution of the design development from a single-band BPF, through a dual-band BPF, to the triband BPF using SICC structures. In the second paragraph, we itemize the improvements of the SICC structure proposed in this paper compared to the SICC structures employed in [10] and [11]. Apparently, the SICC proposed in this paper is more advantageous than those in [10] and [11].

Reviewer 2 Report

1.The manuscript presents a quad-band filter design using substrate-integrated coaxial cavities. It is an interesting job, and is valuable for reducing the size of filter.

2. In the second part Filter Design and Sample Results, "an SMA is used to feed the SICC from the SICC's bottom wall, where the SMA's signal probe penetrates the cavity and reaches the top wall." The reviewer believes the top wall is the metal layer with four embedded fan-shaped patches from Fig.1(a). But author call this top wall as the second metal (fan-shaped patches embedded) from Fig.1(b), and the metal layer with four fan-shaped slots is also called as the top metal from Fig.1(c). At the same time, author said "Four distinct slots, also designed to be of fan shape, are etched on the common top wall for coupling energy between the two stacked SICCs". The reviewer thinks these descriptions may confuse the readers.

Author Response

We thank the reviewer's comments, and we reply to the reviewer's comments in the attached file. The writing is also carefully checked by a pro editor for better quality.

Reviewer 3 Report

This paper mainly gives detailed design quad-band filter with calculation and part experiment information. The results are good for future mobile communication. Questions and comments:

1) How do the authors calculate the frequencies and Q-factors of the filters? A circuit model or commercial codes? How are the accuracy for the calculation?

2) In fig.3, can the author show the electric field with arrow plot instead of cloud-plot? That would be more direct for our readers, and easy to compare and connect with Fig 4 for magnetic field.

Author Response

We thank the reviewer's comments. All the comments are carefully replied to and they are given in the attached file. The writing and English grammar have been carefully checked and corrected by a pro editor for better quality.

Round 2

Reviewer 1 Report

How to control the external quad-mode couplings (or bandwidths) of the four passbands?

Author Response

We are grateful for the reviewer's valuable comments.  The attached file is for replying to the reviewer's comments,  and we wish for the best. 
